# Study of UHMWPE Fiber Surface Modification and the Properties of UHMWPE/Epoxy Composite

**DOI:** 10.3390/polym12030521

**Published:** 2020-03-01

**Authors:** Lei Han, Haifeng Cai, Xu Chen, Cheng Zheng, Weihong Guo

**Affiliations:** Polymer Processing Laboratory, Shanghai Key Laboratory of Advanced Polymeric Materials, Key Laboratory for Ultrafine Materials of Ministry of Education, School of Material Science and Engineering, East China University of Science and Technology, Shanghai 200237, China; 13122320076@163.com (L.H.); 18721358176@163.com (H.C.); chenxu08175056@163.com (X.C.); zhengcheng_ecust@163.com (C.Z.)

**Keywords:** UHMWPE fiber, epoxy resin, PVA, adhesive property, aldol condensation

## Abstract

Ultra-high molecular weight polyethylene (UHMWPE)/epoxy composites with excellent adhesive properties were prepared by forming an interface membrane on the UHMWPE fiber surface. The interface membrane of the UHMWPE fiber and epoxy resin was polymerized by an aldol condensation between polyvinyl alcohol (PVA) and glutaraldehyde. Different surface treatment methods of UHMWPE fibers were optimized and the two-step PVA-glutaraldehyde condensation (Corona-PG-2S) method is the best. The interfacial adhesion between UHMWPE fiber and epoxy resin was enhanced, and the adhesive properties of the composite were improved. X-ray photoelectron spectroscopy (XPS) and energy dispersive spectrum (EDS) results of the fiber treated by Corona-PG-2S shows that the surface oxygen content was up to 25.0 wt %, with an increase of 17.3 wt % compared with the surface oxygen content of unmodified UHMWPE fiber, which indicated that the surface polarity was greatly enhanced. The adhesive properties were improved by improving the polarity of the surface. The peel strength, ultimate cohesive force, tensile strength and flexural strength of the composite treated by Corona-PG-2S were greatly increased to 262.8%, 166.9%, 139.7%, 200.6% compared with those of unmodified samples. The composite prepared by Corona-PG-2S had excellent adhesive properties, demonstrating that the Corona-PG-2S method plays a major role in significantly improving the composite adhesive properties.

## 1. Introduction

Ultra-high molecular weight polyethylene (UHMWPE) fiber has many excellent properties, such as high tensile strength, low specific density, high impact resistance and excellent chemical resistance. It is widely used as bullet material, aerospace material, biomedical material, fishing material, etc [1,2,3]. UHMWPE fiber has an extremely high degree of crystallinity (>99%) and macromolecular orientation (>95%) due to the full extension during the gel spinning procedure, which results in high modulus and tenacity of UHMWPE fiber [4,5,6]. UHMWPE fiber-reinforced composite has received great attention due to its outstanding properties. However, the simple methylene structure leads to an inert chemical surface, which makes the fiber difficult to react with epoxy resin. In addition, highly oriented UHMWPE fiber is not easily permeated by epoxy resin and it has poor hydrophilicity, which limits the wide application of UHMWPE fiber. Therefore, improving the adhesive properties of UHMWPE fiber has become one of the topics of great concern.

At present, there are many modification methods of UHMWPE fiber [7,8,9,10,11], such as: plasma treatment [12,13,14,15,16], UV-initiated grafting [17], electron beam irradiation-induced grafting [18], X-ray irradiation grafting [19,20] and so on [21]. Corona treatment technology has been widely used to modify the surface of UHMWPE fiber for decades. At the same time, it is an environmentally friendly process and an effective way to improve polymer surface adhesion by introducing polar groups and increasing the surface roughness without affecting the bulk properties [22,23,24,25].

Rodrigues used low-pressure RF plasma to explore the influence of factors such as treatment power, treatment time and oxygen content [26]. Shi introduced hydrophilic groups on the surface of UHMWPE fiber by atmospheric pressure argon plasma treatment [27]. They proved that this method had a good effect on increasing the durability. In order to improve the adhesive properties of UHMWPE fiber, Ren Yu modified the UHMWPE fiber by air dielectric barrier discharge plasma and great results were achieved [28]. The test showed that the adhesive property of the UHMWPE fiber was improved from 1.27 MPa to 2.10 MPa after modification, with an increased rate of 65%. Xing Zhe et al. grafted methyl acrylate (MA) monomer on the surface of UHMWPE fiber by γ-ray pre-irradiation. After a 2 h reaction, at the condition of 50 kGy, 60 °C, 80% monomer volume fraction, the grafting rate reached 190.4%. After grafting, the surface of UHMWPE fiber was rough and oxygen-containing functional groups were introduced.

Compared with X-ray irradiation grafting, the Corona-PG-2S method is safer and easier to commercialize. Compared with plasma treatment, the Corona-PG-2S method can further improve the content of oxygen groups on the surface, thus the adhesive properties can be further improved. In this research, a simple and effective method of two-step surface treatment of UHMWPE fiber treated by corona (Corona-PG-2S) was proposed. The adhesive properties between UHMWPE fiber and epoxy resin were improved due to the introduced oxygen groups on the fiber surface through the corona. However, the oxygen groups on the surface of the composite treated by corona were relatively fewer, therefore, an effective modification method should be proposed. Based on corona treatment, a two-step PVA-glutaraldehyde condensation method was proposed to increase the oxygen groups. The corona treated UHMWPE fiber reacted with PVA firstly, and then polymerized with glutaraldehyde. Finally, a polarized membrane with many oxygen-containing groups was formed on the UHMWPE fiber surface. The composite prepared by Corona-PG-2S had excellent adhesive properties, thus the Corona-PG-2S method played an important role in improving the properties of the composite significantly.

## 2. Experimental

### 2.1. Materials and Instruments

#### 2.1.1. Materials

UHMWPE fiber and UHMWPE fiber cloth were provided by Honeywell Integrated Technology Co., Ltd (Shanghai, China). Glutaraldehyde and Polyvinyl alcohol were purchased from Shanghai Aladdin Biochemical Technology Co., Ltd (Shanghai, China). Epoxy resin (E-44), silicone oil cleaning agent, curing agent (synthesis of diethylenetriamine and propylene oxide butyl ether) were purchased from Kunshan South Asia Epoxy Resin Co., Ltd (Jiangsu, China) [29]. Table 1 is the data of the physical properties of UHMWPE fiber.

#### 2.1.2. Instruments

Corona processor (Honeywell Integrated Technology Co., Ltd, Shanghai, China) was used to treat UHMWPE fiber. The corona voltage was 100 V and the corona current was 2.40 A, lasting for 1 min.

Flat vulcanizer (Hua Kang Co., Ltd, Shanghai, China) was used to prepare the UHMWPE/epoxy composite. The vulcanization temperature was 20 °C and the vulcanization pressure was 10 MPa, lasting for 8 h.

### 2.2. Modification of UHMWPE Fiber

Figure 1 is the mechanism of an aldol condensation. The reaction between PVA and glutaraldehyde generates hemiacetal firstly, and then hemiacetal continues to generate acetal. As a result, the surface of the formed membrane contains a large number of ether bonds. Therefore, the adhesive properties of the UHMWPE fiber were greatly improved by aldol condensation method. Compared with the method of corona, Corona-PG-2S forms a protective membrane on the UHMWPE fiber surface, which reduces the damage to the UHMWPE fiber surface and increases oxygen-containing groups. Therefore, the adhesive properties are significantly improved. In order to further explore the influencing factors of two-step PVA-glutaraldehyde condensation, one-step PVA-glutaraldehyde condensation, and corona, surface treatment methods of the UHMWPE fiber surface are listed in Table 2.

### 2.3. Preparation of UHMWPE/Epoxy Composite

Figure 2 is the preparation process of the composite. In order to facilitate the sample preparation process, the fiber was replaced by fiber cloth to prepare the UHMWPE/epoxy composite. The UHMWPE fiber cloth was treated by H_2_SO_4_ and PVA, glutaraldehyde solution respectively, then epoxy resin and the curing agent were thoroughly mixed and uniformly coated on the dried UHMWPE fiber cloth. The composite was pressed by the flat vulcanizer. The vulcanization temperature was 20 °C and the vulcanization pressure was 10 MPa, lasting for 8 h. A rectangular sheet with a thickness of 2 mm was molded and cut into strips for testing. Table 3 is the mass fraction of epoxy, curing agent, the fiber in the UHMWPE/epoxy composite.

Figure 3 is the mechanism of Corona-PG-2S of UHMWPE fiber [30]. Figure 3 shows that some oxygen-containing groups can be introduced on the surface of the UHMWPE fiber when the fiber treated by corona, including some –OH groups and a few C=O groups. However, the oxygen-containing groups are relatively few and unevenly distributed and the corona does certain damages to the surface of the UHMWPE fiber, thus the improvement of adhesive properties of the UHMWPE/epoxy composite is not obvious.

After treated by corona, the UHMWPE fiber was soaked in the aqueous solution of H_2_SO_4_ and PVA. During the process of reacting with PVA, a new layer of PVA on the fiber surface was formed. The UHMWPE fiber treated by corona and PVA has more –OH groups than that treated by corona. The third step was to soak the PVA coated UHMWPE fiber in the glutaraldehyde solution, and then the aldol condensation reaction occurs. Finally, a new membrane with C–O–C groups on the UHMWPE fiber surface was prepared. The introduced C–O–C groups are beneficial for the adhesive properties of the UHMWPE/epoxy composite. Therefore, UHMWPE fiber treated by the Corona-PG-2S has better adhesive properties than the unmodified fiber and the fiber treated by corona.

### 2.4. Characterization

#### 2.4.1. Characterization of UHMWPE Fiber

The Fourier transform infrared (FTIR) spectra were recorded on a Nicolet 6700 FTIR spectrometer (New York, NY, USA) at a scan number of 32 and a resolution of 4 cm^−1^. The surface morphology was observed using an S-4800 field emission scanning electron microscope (SEM, Tokyo, Japan). All samples were coated with a thin gold layer before SEM observation, and the element content on the surface was calculated by using an energy dispersive spectrum (EDS) system (QUANTAX 400-30, New York, NY, USA) and X-ray photoelectron spectroscopy (XPS, ESCALAB 250Xi, New York, NY, USA) with an Mg Kαphoton energy of 1253.6 eV.

#### 2.4.2. Characterization of UHMWPE/Epoxy Composite

The peel strength, ultimate cohesive force, tensile strength and bending strength of the UHMWPE/epoxy composite were tested by using an MTS E44 testing machine in accordance with ISO 4578, ISO 527 and ISO 178, respectively [30]. At least five independent measurements were conducted for each sample (80 × 10 × 2 mm for tensile and ultimate cohesive force test, 50 × 10 × 2 mm for peel strength test). The bending properties were tested in a three-point bending mode. At least five independent measurements were conducted for each sample (60 × 10 × 2 mm for bending strength test).

## 3. Results and Discussion

### 3.1. Properties of UHMWPE/Epoxy Composite

Figure 4 is the peel strength of the UHMWPE/epoxy composite with different PVA concentrations (The results of ultimate cohesive force, tensile strength and bending strength tests can be found in Appendix A). In order to make the experiment more convincing, we also prepared and tested the UHMWPE fiber/epoxy composite treated by Uncorona-PG-1S, Uncorona-PG-2S and Corona-PG-1S. Figure 4 shows that six modification methods have different effects on the adhesive properties of the UHMWPE/epoxy composite. F-f in Figure 4 shows that Corona-PG-2S is the best method among these methods. The UHMWPE/epoxy composite treated by corona has higher adhesive properties than the unmodified composite, this is because more –OH groups were introduced on the surface of the fiber after treated by corona.

The methods F-a, F-b and F-f in Figure 4 show that the peel strength of the composite treated by Corona-PG-2S is 2.01 N/mm, with an increase of 262.82% compared with that of unmodified UHMWPE fiber and 30.52 % compared with that of fiber UHMWPE treated by corona, respectively.

The comparison result between the method F-d and F-f in Figure 4 shows that method F-f has higher peel strength, the reasons are: 1) the fiber treated by corona can react with –OH groups of PVA firstly, therefore PVA can be adsorbed on the surface of the fiber better; 2) the immediate materials can react with glutaraldehyde to form a uniform membrane on the fiber surface. If the fiber isn’t treated by corona, PVA will only attach on the surface of the fiber and there will not be a chemical bond. As a result, the formed membrane is loose when reacting with glutaraldehyde. The loose membrane will result in poor adhesive properties of the composite.

The comparison result between the method F-e and F-f in Figure 4 shows that the fiber treated by two-step PVA-glutaraldehyde condensation has higher peel strength than that of one-step method. This is because when PVA and glutaraldehyde were added into the beaker at the same time in a one-step method, some PVA and glutaraldehyde will react immediately before the PVA attaches on the UHMWPE fiber surface, which affects the formation of the membrane. The two-step PVA-glutaraldehyde condensation can ensure that PVA is evenly distributed on the surface of the fiber firstly, and then the PVA on the surface of the fiber reacts with glutaraldehyde, forming a dense membrane on the UHMWPE fiber surface. The membrane so formed is tighter and more compact. Therefore, the two-step method has higher adhesive properties than one-step method.

Figure 4 shows the UHMWPE/epoxy composite has the highest peel strength when the PVA concentration is about 0.1 g/mL. Figure 5 is the peel strength of the smaller PVA concentration gradient of Corona-PG-2S. Figure 5 shows that the peel strength is the highest when the PVA concentration is 0.10 g/mL. The peel strength of UHMWPE fiber is improved with the increase of PVA concentration firstly, and the peel strength reaches the best when the PVA concentration is 0.10 g/mL, with the increase of PVA concentration, the peel strength of the composite became poor.

Figure 4 and Figure 5 show that the six methods have similar curves and all the methods have their highest peel strength when the PVA concentration is 0.10 g/mL. This is because when the PVA concentration is more than 0.10 g/mL, PVA will aggregate on the surface of the UHMWPE fiber, which results in an uneven distribution of PVA, thus the formed membrane is loose. When the concentration of PVA is less than 0.10 g/mL, the content of PVA is too low to form a complete membrane structure on UHMWPE fiber. The similar curves of the six methods prove that the most suitable concentration for forming a dense membrane structure is 0.10 g/mL. Incomplete membrane structure makes the fiber is not completely covered, which leads to relatively few oxygen groups of the surface, as a result, the membrane is also easy to separate from the fiber. The loose membrane leads to a poor combination between the fiber and resin. Therefore, both the loose membrane and incomplete membrane can lead to the poor peel strength, and the peel strength is the best when the concentration is 0.10 g/mL.

Figure 6 is the typical stress–strain curve of the composite of the surface treatment methods. Figure 6 shows that the UHMWPE fiber treated by Corona-PG-2S has the best tensile strength, as a result, the tensile strength is 258.91 Mpa. Because the method of Corona-PG-2S is the best modification method we have explored, this article will research the properties of unmodified fiber, fiber treated by corona and fiber treated by Corona-PG-2S in the later exploration.

### 3.2. Properties of UHMWPE Fiber

#### 3.2.1. Surface Morphology

Figure 7 is the surface morphology of unmodified UHMWPE fiber, UHMWPE fiber treated by corona and UHMWPE fiber treated by Corona-PG-2S at 0.10 g/mL PVA concentration. Method F-a in Figure 7 shows that the surface of unmodified UHMWPE fiber is rough, but the surface is smoother than that of UHMWPE fiber treated by corona. The method F-b in Figure 7 shows the surface of the fiber treated by corona is extremely rough. This is because the UHMWPE fiber treated by corona was etched and mechanical meshing points on the surface were much more than unmodified fiber. The meshing point means the increase of the fiber surface roughness, thus the adhesive properties between the fiber and epoxy can be improved. Figure 7 shows that the surface of UHMWPE fiber treated by Corona-PG-2S is the smoothest, and a dense membrane is formed on the surface of the fiber. Although there are no mechanical meshing points, the formed membrane contains lots of oxygen groups on the fiber surface. As a result, the adhesive properties of UHMWPE fiber treated by Corona-PG-2S are better than these properties of the fiber treated by corona and unmodified fiber.

#### 3.2.2. Functional Groups

Figure 8 is functional groups of FTIR spectra of the fiber treated by method F-a, F-b and F-f at 0.10 g/mL PVA. In Figure 8, the broad bands at 2918 cm^−1^ and 2850 cm^−1^ belong to the Asymmetric and Symmetric Stretching Absorption of –CH_2_–, respectively; the broad bands at 1465 cm^−1^ and 720 cm^−1^ belong to the bending and rocking deformation absorption of –CH_2_–, respectively [31]. Compared with unmodified UHMWPE fiber, three new absorption peaks are observed at around 1715 cm^−1^, 1175 cm^−1^ and 1050 cm^−1^ in method F-b and method F-f, which belongs to the characteristic absorption of the C=O [32], C–O–C and –OH, respectively.

The method F-b in Figure 8 shows that the –OH, C=O and C–O–C groups have been successfully introduced on the surface of fiber treated by corona. The method F-f in Figure 8 shows that the –OH, C=O and C–O–C groups have been successfully introduced on the surface of fiber treated by Corona-PG-2S. The method F-f shows the C–O–C groups were formed on the fiber. This is because the –OH groups of PVA can react with the glutaraldehyde to form C–O–C groups in the process of the aldol condensation reaction. As a result, the method F-f in Figure 8 shows that C–O–C groups are formed on the UHMWPE fiber surface.

#### 3.2.3. Fiber Surface Element Content Analysis of EDS

Table 4 is the EDS diagram of the carbon and oxygen content of UHMWPE fiber surface. The measurement spectrum provides a qualitative description of the elements of the fiber surface. Table 4 shows that the oxygen content of the unmodified fiber surface is 7.69 wt %, while the oxygen content of the fiber surface treated by corona and Corona-PG-2S is 18.18 wt % and 25.07 wt %, respectively. Compared with the fiber treated by corona, the oxygen content of fiber treated by Corona-PG-2S increased by 6.89 %. The EDS result shows that the oxygen content of the fiber surface treated by Corona-PG-2S is higher than that of the fiber surface treated by corona.

#### 3.2.4. Fiber Surface Element Content Analysis of XPS Spectra

Figure 9 shows the UHMWPE fiber surface element analysis of the XPS spectra of different surface treatment methods. Figure 9 shows the chemical composition of the fiber surface and it proves that there are only carbon and oxygen elements on the fiber surface. The measurement spectrum provides a qualitative description of elements of the fiber surface, and a qualitative description of the high-resolution spectra of O1S and C1S, which determines the chemical state and quantifies the compositions.

Figure 10A,B shows the XPS diagrams of oxygen and carbon content of the unmodified fiber. From Figure 10A,B, the ratio of carbon content and oxygen content is 12:1. This is because the surface of the unmodified fiber is formed by CH_2_ straight-chain structure, and the oxygen content of the fiber surface is very low, thus the carbon content of the fiber surface is much higher than the oxygen content. Figure 10A shows one peak of the oxygen element. The peak at 531.8 eV belongs to –OH. Figure 10B shows two peaks of carbon element. The peaks at 284.6 eV and 285.0 eV belong to C–C and C–OH. Figure 10A,B proves that there are few –OH and no C–O–C on the surface of the unmodified fiber, indicating that the adhesive properties of the fiber are poor.

Figure 11A,B shows the XPS diagrams of oxygen and carbon content of the fiber treated by corona. Figure 11A,B shows the ratio of carbon content and oxygen content is 4.5:1. This is because when the fiber is treated by corona, some oxygen-containing groups are formed on the surface of the fiber.

Figure 11A shows two peaks of oxygen element, the peaks at 531.8 eV and 533.3 eV belong to –OH and C=O, there are few C–O–C groups. However, the content of C=O is very low, and the –OH is the main functional group of the fiber surface treated by corona. Figure 11B shows three peaks of carbon element, the peaks at 284.6 eV, 285.0 eV and 287.5 eV belong to C–C, C–OH and C=O [33]. Figure 11A,B proves that the amount of –OH and C=O increases after the fiber is treated by corona, thus the method of corona improves the adhesive properties of UHMWPE fiber.

Figure 12A,B is the XPS diagrams of oxygen and carbon content of the fiber treated by Corona-PG-2S. Figure 12A,B shows the ratio of carbon content and oxygen content is 3:1, and the oxygen content is further increased. Figure 12A shows two peaks of oxygen element. The peaks at 531.8 eV and 532.7 eV belong to –OH and C–O–C. The C–O–C contains 74.6% of the oxygen-containing groups, thus the content of C–O–C increases a lot, because C–O–C is formed during the aldol condensation process. Figure 12B shows three peaks of carbon element, the peaks at 284.6 eV, 285.0 eV and 288.7 eV belong to C–C, C–OH and C–O–C. Figure 12A,B proves that C–O–C of the fiber surface increases a lot after it is treated by Corona-PG-2S, thus the fiber treated by Corona-PG-2S has the best adhesive properties.

Figure 12 proves that the oxygen content of the fiber treated by Corona-PG-2S is higher than that of the fiber treated by corona. This is because during the first step, a new reaction occurs between –OH of PVA and the –OH of the fiber treated by corona, thus more –OH of PVA is introduced on the fiber surface. The –OH of the fiber surface reacts with glutaraldehyde in the second step. Therefore, the fiber treated by Corone-PG-2S has more oxygen content than the fiber treated by corona, which indicates that the fiber treated by Corone-PG-2S has better adhesive properties.

## 4. Conclusions

In this study, the two-step PVA-glutaraldehyde condensation method (Corona-PG-2S) was proposed to improve the adhesive properties of the fiber. The corona method can increase the oxygen groups of the fiber surface, but the effect is not obvious. Compared with the corona method, the proposed Corona-PG-2S method can form a membrane with more oxygen groups by PVA-glutaraldehyde condensation on the basis of the corona. Compared with the fiber treated by corona, the oxygen content of the fiber surface increased 6.9 wt % and the C–O–C content of the fiber surface increased significantly. The result of SEM shows that a smooth membrane was formed on the surface of the UHMWPE fiber. The mechanical result shows that the membrane can tightly wrap the fiber when the PVA concentration is 0.10 g/mL. The peel strength, ultimate cohesive force, tensile strength and flexural strength of the composite treated by Corona-PG-2S were 2.01 N/mm, 24.09 N, 258.91 MPa, 58.29 MPa, with an increase of 30.52%, 26.59%, 32.82%, 32.42% compared with these properties of the fiber treated by corona. Therefore, in this study, compared with the corona method, the Corona-PG-2S method increased the content of oxygen groups and it has the best adhesive properties.

## Figures and Tables

**Figure 1 polymers-12-00521-f001:**
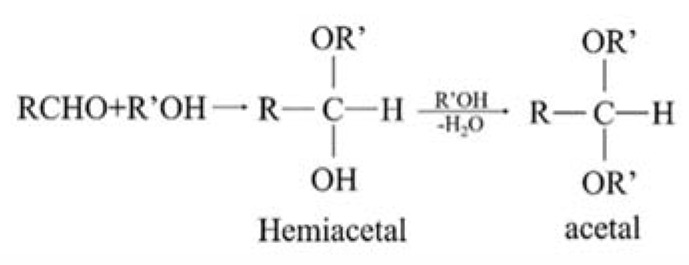
The mechanism of an aldol condensation.

**Figure 2 polymers-12-00521-f002:**
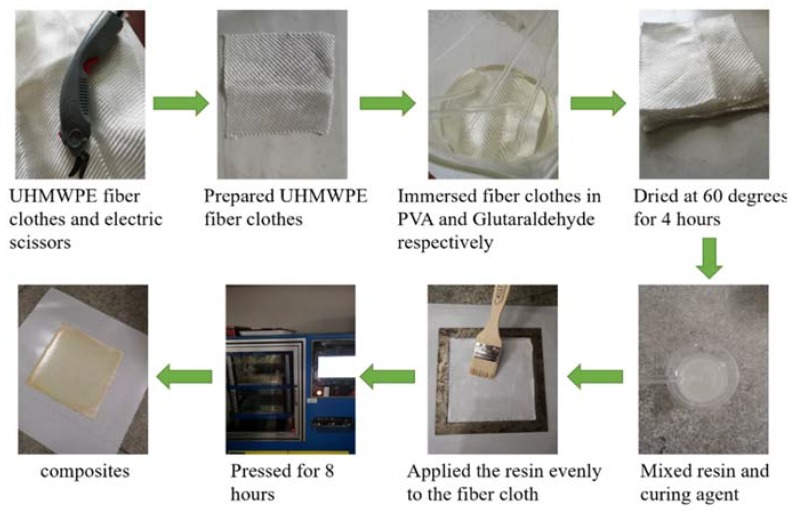
Preparation of the UHMWPE/epoxy composite.

**Figure 3 polymers-12-00521-f003:**
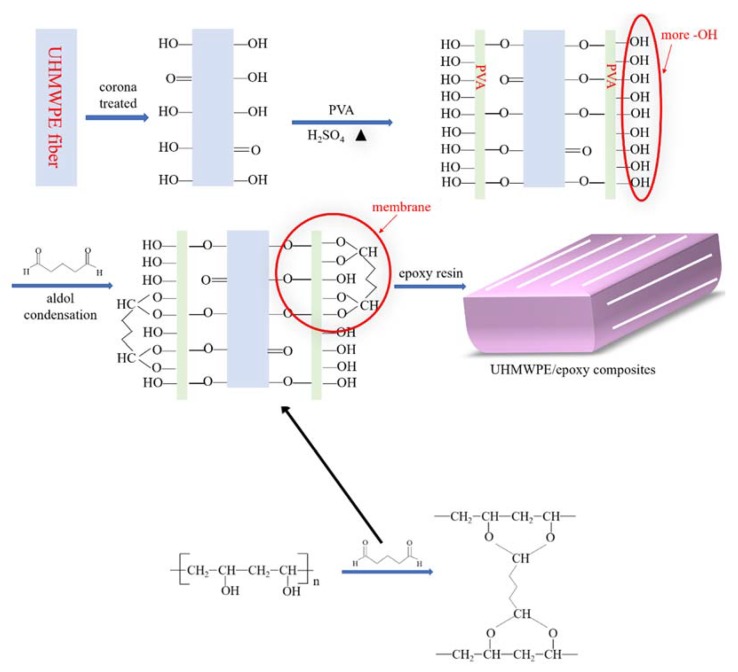
Mechanism of the UHMWPE fiber treated by Corona-PG-2S.

**Figure 4 polymers-12-00521-f004:**
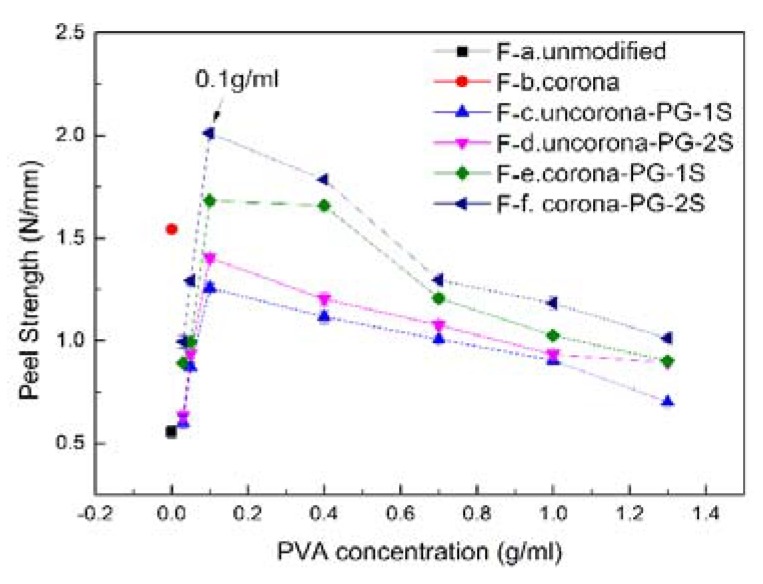
Peel Strength of the composite with different polyvinyl alcohol (PVA) concentration of the six methods.

**Figure 5 polymers-12-00521-f005:**
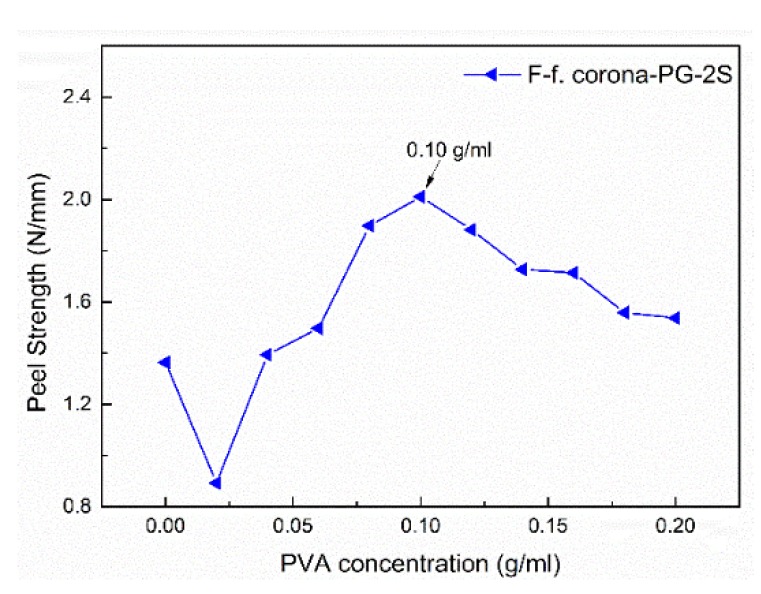
Peel Strength of the composite treated by the Corona-PG-2S with smaller PVA concentration gradient.

**Figure 6 polymers-12-00521-f006:**
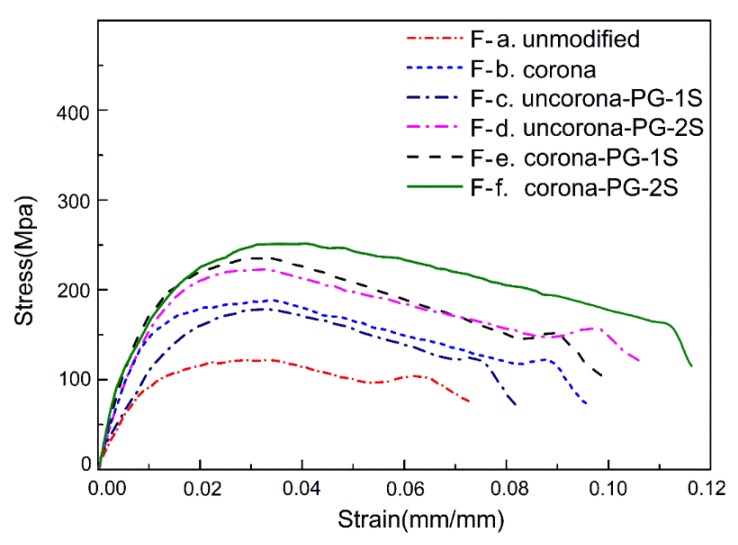
Typical stress–strain curve of the composite of surface treatment methods.

**Figure 7 polymers-12-00521-f007:**
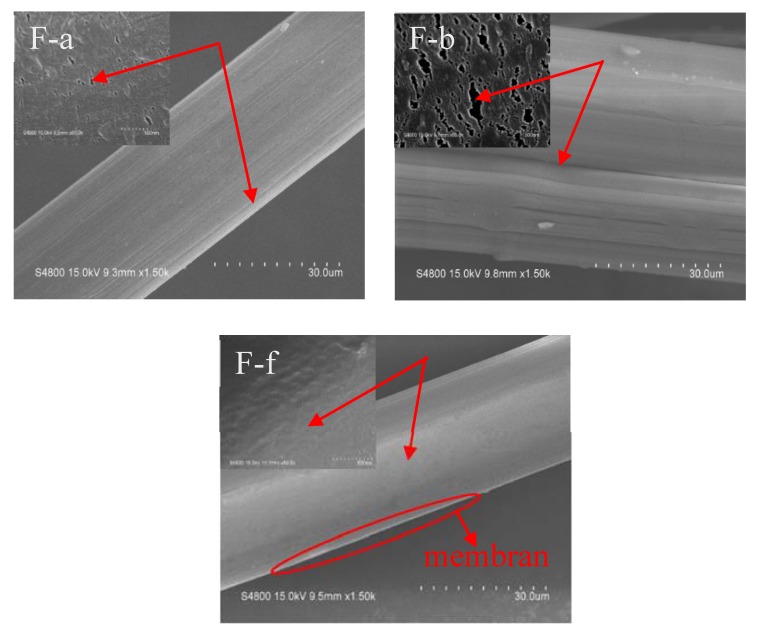
Surface morphology of the fiber treated by method F-a, F-b and F-f at 0.10 g/mL PVA.

**Figure 8 polymers-12-00521-f008:**
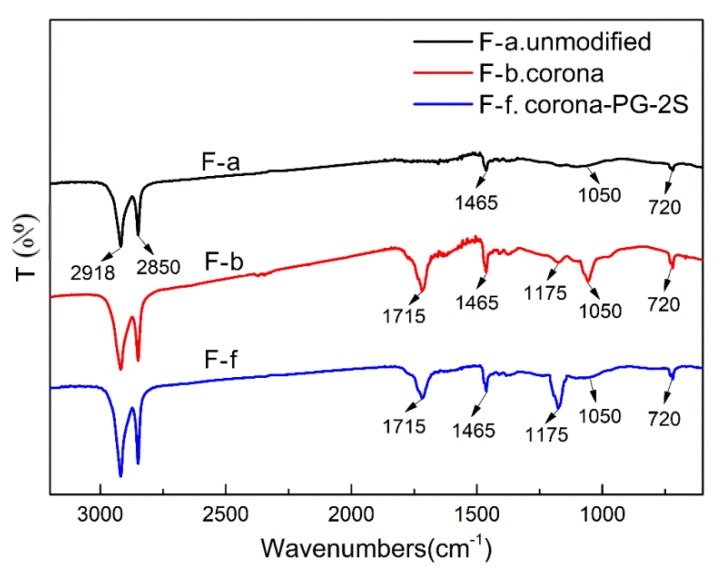
Functional groups of FTIR spectra of the fiber treated by method F-a, F-b and F-f at 0.10 g/mL PVA.

**Figure 9 polymers-12-00521-f009:**
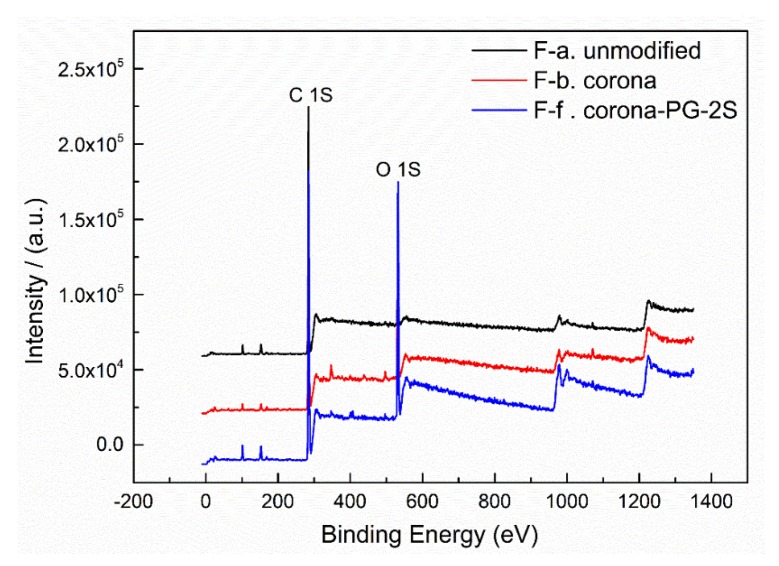
Fiber surface element analysis of X-ray photoelectron spectroscopy (XPS) spectra of the fiber treated by method F-a, F-b and F-f at 0.10 g/mL PVA.

**Figure 10 polymers-12-00521-f010:**
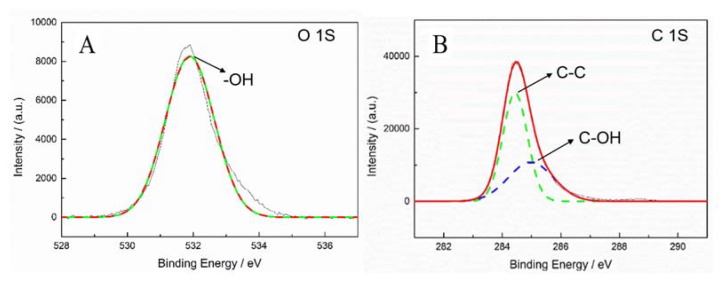
Fiber surface element analysis of XPS spectra of the fiber treated by method F-a. (**A**) O 1S, (**B**) C 1S.

**Figure 11 polymers-12-00521-f011:**
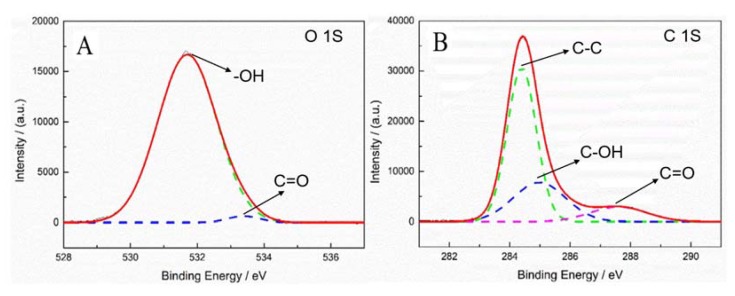
Fiber surface element analysis of XPS spectra of the fiber treated by method F-b. (**A**) O 1S, (**B**) C 1S.

**Figure 12 polymers-12-00521-f012:**
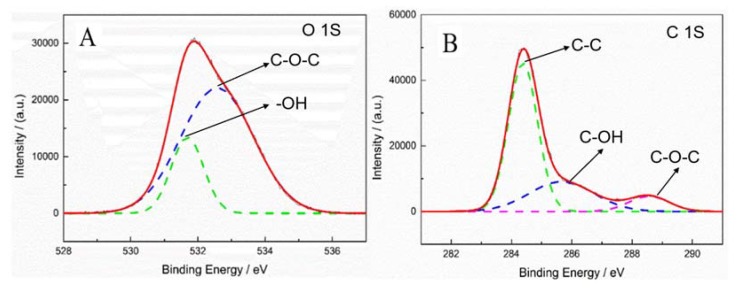
Fiber surface element analysis of XPS spectra of the fiber treated by method F-f. (**A**) O 1S, (**B**) C 1S.

**Table 1 polymers-12-00521-t001:** Physical properties of the Ultra-high molecular weight polyethylene (UHMWPE) fiber.

Molecular Weight	Density (g/cm^3^)	Elastic Modulus (MPa)	Water Absorption (%)
5.3 × 10^6^	0.927–0.941	600	<0.01

**Table 2 polymers-12-00521-t002:** Surface treatment methods of the UHMWPE fiber.

Modification	Abbreviation						
	F-a	F-b	F-c	F-d	F-e	F-f
Corona	N	Y	N	N	Y	Y
PG-1S	N	N	Y	N	Y	N
PG-2S	N	N	N	Y	N	Y

F-a: Unmodified UHMWPE fiber; F-b: UHMWPE fiber treated by corona; F-c: UHMWPE fiber treated by the method of Uncorona-PG-1S; F-d: UHMWPE fiber treated by the method of Uncorona-PG-2S; F-e: UHMWPE fiber treated by the method of Corona-PG-1S; F-f: UHMWPE fiber treated by the method of Corona-PG-2S; Corona: UHMWPE fiber treated by corona for 1 min, the conditions were: 100V power supply voltage, 2.4 A power supply current; PG-1S: The fiber was soaked in H_2_SO_4_ and PVA and glutaraldehyde solution for 1 h and dried at 60 °C for 4 h; PG-2S: The fiber was soaked in H_2_SO_4_ and PVA solution for 0.5 h and in glutaraldehyde solution for 0.5 h and dried at 60 °C for 4 h.

**Table 3 polymers-12-00521-t003:** Mass fraction of epoxy, curing agent, the fiber in UHMWPE/epoxy composite.

Element	Sample						
		F-a	F-b	F-c	F-d	F-e	F-f
Epoxy Resin	Weight (wt %)	35 ± 0.5	35 ± 0.5	35 ± 0.5	35 ± 0.5	35 ± 0.5	35 ± 0.5
Curing Agent	Weight (wt %)	5 ± 0.2	5 ± 0.2	5 ± 0.2	5 ± 0.2	5 ± 0.2	5 ± 0.2
Dried UHMWPE	Weight (wt %)	60 ± 1.0	60 ± 1.0	60 ± 1.0	60 ± 1.0	60 ± 1.0	60 ± 1.0

**Table 4 polymers-12-00521-t004:** Mass fraction of carbon and oxygen on the UHMWPE fiber surface.

Element	Unmodified	Corona	Corona-PG-2S
	[wt %]	[wt %]	[wt %]
Carbon	92.31	81.82	74.93
Oxygen	7.69	18.18	25.07

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
