# Peer review of "Study of UHMWPE Fiber Surface Modification and the Properties of UHMWPE/Epoxy Composite"

_polymers, 2020, doi:10.3390/polym12030521_

Round 1

Reviewer 1 Report

Several improvements are necessary before publications:

1) Figure1 Mechanism of aldol condensation. and Figure 2 Preparation of UHMWPE/epoxy composite could be placed in Suplimentary, because they do not provide any scientific information.

2) Please provide motivation (cons and pros) in the introduction or other section of the text why authors have chosen to use Corona and two-step PVA-glutaraldehyde condensation modification for UHMWPE textile. It need to be compared with other existing modification methods.

3) Figure 4 A-D and Figure 5A-D are very similar in changes nature; It is advised to show only data for single test; other similar graphics to put in suplimentary or even skipped. This discussion section need to be enhanced strongly - why the curves looks so similar; what is the mechanism about it.

4) Several misleading sentences need to be corrected in the text. For example : 

The method F-b shows that the -OH, C=O and C-O-C groups have been successfully introduced

245  on the surface of fiber treated by corona,-OH groups and C=O groups are much more than C-O-C

246  groups.

How it is concluded that OH and CO groups are much more than COC ? Please read the text for similar sentences.5) 

5) What information provides Fig. 9 ? It is not clear for the reader. Please improve this section.

6) The conclusion need to compare two modification methods and conclude about their efficiency.

Reviewer 2 Report

In my opinion, the article: “Study on UHMWPE fiber surface modification and 3 the properties of UHMWPE/epoxy composite” is interesting. The investigation has been correctly carried out from an experimental point of view and the obtained results and outcomes have been accurately discussed. From the purpose, also the experimental part has been sufficiently described.

Unfortunately, the manuscript has to be significantly improved from the English language. For the purpose, the authors, before to propose again the manuscript, are recommended to provide for a mother tongue revision.

Here some questions:

1 – Raws 11-12. The authors claim: “Different surface treatment methods of
UHMWPE fiber optimized”.

- It seems that there is some missing verbs or words;

Pag 1. Introduction section: In my opinion the English language has to be improved;

Pag 2 – Raws 50-52 - The authors claim: “The test showed that the adhesive properties of the UHMWPE fiber surface increase from 1.27 MPa before modification to 2.10 MPa at the highest, with an increase rate of 65%”.

- The authors are invited to rewrite the sentence with an improved English language;

Pag 2 – Raws 64-66. The authors claim: “The excellent adhesive properties
of composite prepared by Corona-PG-2S demonstrating that the method of Corona-PG-2S playing an important role in such significant improvements in composite properties”.

- The authors are invited to rewrite the sentence with an improved English language;

Page 3 – raw 113. The authors are invited to improve the process description, mainly in the English language;

Moreover, they are invited to specify process conditions, as temperature and pressures, set up in the pressing phase;

The authors uses many continuously the word “immerse” in the experimental phase. They are invited to replace it with something else, as, i. e., “Dip”;

Pag 6 – Raws 177-182. The authors claim: “The comparison of the method F-c and F-e, the method F-d and F-f in Fig. 4, shows that method F-e and F-f has higher adhesive properties, the reasons are: 1) the fiber treated by corona can react with -OH groups of PVA firstly, therefore PVA can be adsorbed on the surface of the fiber better; 2) the immediate materials can react with glutaraldehyde to form a uniform membrane on the surface of fiber. The fiber without treated by corona, PVA is only attached to the surface of the fiber, and there is no chemical bond”

- The authors are invited to improve the English language to increase the sentence readability;

Pag 6 – Raws 191. The authors claim: “As a result, the membrane formed is tighter and more compacter,..”

- The authors are invited to replace the term “compacter” with “compact” or something else;

7 – Raws 202-203. The authors claim: “When the concentration of PVA is less than 0.10g/ml, the content of PVA is too few to form a complete membrane structure on UHMWPE fiber.”.

- The authors are invited to replace as it follows: “When the concentration of PVA is less than 0.10g/ml, the content of PVA is too low to form a complete membrane structure on UHMWPE fiber.”;

9 – Raws 225-226. The authors are invited to specify the meaning of “Meshing Point”;

12 – Raws 273-277. The authors claim: “Fig. 10A and Fig. 10B are the XPS diagrams of oxygen and carbon content of the unmodified UHMWPE fiber. From Fig. 10A and Fig. 10B, the ratio of carbon element and oxygen element is 12:1. This is because the surface of unmodified UHMWPE fiber is CH2 straight chain structure, and the content of oxygen groups on the surface are very few, therefore the content of carbon element on the
surface of the UHMWPE fiber is far more than the content of O”.

- The authors are invited to improve the English language to increase the sentence readability;

Round 2

Reviewer 1 Report

The paper can be accepted for publishing

Author Response

Thank you for your advice, it's very useful for me!